# A Predictive Model to Evaluate the HbeAg Positivity of Chronic Hepatitis B Virus Patients in Clinics: A Cross-Sectional Study

**DOI:** 10.3390/medicina58091279

**Published:** 2022-09-15

**Authors:** Ning Wang, Jinli Zheng, Yang Huang, Xingyu Pu, Li Jiang, Jiayin Yang

**Affiliations:** Department of Liver Surgery, Liver Transplantation Center, West China Hospital of Sichuan University, Chengdu 610000, China

**Keywords:** HBeAg-positive, HBeAg-negative, serum HBV DNA levels, chronic hepatitis B, correlation

## Abstract

*Background and Objective*: The study aims to investigate the correlation between Hepatitis B ‘e’ antigen (HBeAg) and HBV DNA levels, and to find a convenient tool to estimate the HBV DNA level for clinicians. *Materials and Methods*: We enrolled 1020 patients in this cross-sectional study and divided them into four groups: an HbeAg-positive and -negative group, and high and low HBV DNA levels groups. *Results*: Alanine aminotransferase (ALT), Albumin (ALB) and HBeAg are independent risk factors for CHB patients. When the level of HBeAg is higher than 16.15 S/CO, it is four times more likely that the patients will have high levels of HBV DNA than those who do not. The ALT and TB are independent risk factors in HBeAg-negative patients with a high HBV DNA level. We have drawn three predictive models to estimate the HBV DNA levels for those with the chronic hepatitis B virus (CHB), and those that are HBeAg-positive and HBeAg-negative (Y_1_ = 0.004 × ALT(IU/L) + 1.412 × HBeAg (S/CO) − 0.029 × ALB (g/L) + 0.779, the AUC is 0.672, and the cutoff value is −0.072, there the sensitivity is 0.615, the specificity is 0.648, PPV is 65.182% and NPV is 60.837%; Y_2_ = 0.007 × HBeAg (S/CO) − 0.016 × HGB (g/L) + 3.070, the AUC is 0.724, and the cutoff value is 1.216, where the sensitivity is 0.626, the specificity is 0.897, PPV is 94.118% and NPV is 34.437%; Y_3_ = −0.005 × ALT(IU/L) + 0.006 × TB (umol/L) + 0.385, the AUC is 0.661, and the cutoff value is 0.263, where the sensitivity is 0.677, the specificity is 0.587, PPV is 66.820% and NPV is 40.774%, respectively). We propose that HBeAg is the most important risk factor for the patient with a high HBV DNA level, however, it is not as important in the HBeAg-positive group. *Conclusions*: HBeAg is an independent risk factor that reflects the level of HBV DNA with a strong correlation. Patient with HBeAg (−) should combine TB and ALT to estimate the level of HBV DNA.

## 1. Introduction

The hepatitis B virus (HBV) infection is a serious public health problem worldwide. The World Health Organization has reported that 296 million people would be infected with chronic hepatitis in 2019, with 1.5 million new infections each year, and 820,000 people who have died from HBV infection-related diseases [1]. More importantly, HBV infection may cause liver diseases such as acute or chronic hepatitis, cirrhosis, liver decompensation, and hepatocellular carcinoma [2]. On the other hand, the replication of HBV is the main reason for the development of acute or chronic hepatitis. In addition, previous studies have reported that the depression of the activity of HBV could retard the progress of cirrhosis and reduce the rates of acute or chronic hepatitis [3,4]. Generally, the natural courses of CHB include several phases which are as follows: (I) The immune tolerance phase (IT), with hepatitis B e antigen positivity (HBeAg (+)), high HBV-DNA levels, and normal levels of alanine aminotransferase (ALT). (II) The immune clearance phase (IC), with HBeAg (+), high HBV-DNA levels, normal or high ALT levels. (III) The low-replicative phase (LR), with HBeAg-negativity (HBeAg (−)), hepatitis B e antibody positivity (HbeAb (+)), undetectable levels of HBV DNA and normal ALT levels. (IV) HBeAg-negative hepatitis (ENH), with HBeAg (−), HbeAb (+), high HBV-DNA and ALT levels [5,6,7,8]. HBV DNA is a risk factor for liver cirrhosis as Iloeje UH and Yang HI et al. [9] have reported, where they found that along with HBV DNA with the levels of 10^4^–10^5^ copies/mL (2000–20,000 IU/mL), 10^5^–<10^6^ copies/mL (20,000–200,000 IU/mL) and >10^6^ copies/mL (200,000 IU/mL), the risks of the patient developing liver cirrhosis are 2.5, 5.6 and 6.5 folds, respectively. Therefore, it is meaningful to detect the level of HBV-DNA for the CHB, regularly.

The detection of HBV DNA levels is a common method for assessing the impact of treatment regimens on patients and the efficacy of antiviral therapy [10,11]. Existing studies have reported a correlation between HBsAg and HBV DNA levels, and they have suggested that serum HBsAg levels can serve as a marker for the quantitative prediction of HBV DNA levels [12,13,14,15]. However, similar studies have shown no correlation between HBsAg and HBV DNA [16,17], although the previous study has been used to quantitatively reflect HBV DNA levels by relatively accurately measuring the serum HBsAg levels. According to the study by Gupta E and Kumar A et al. [10], the optimal cut-off value of serum-HBsAg content for predicting a high HBV DNA level is 3.36 × 103 IU/mL, meanwhile, it is difficult to identify the quantification of serum HBeAg in clinics, thus limiting its application. On the other hand, several studies have reported that the relationship between these two parameters were weak [6,16]. In contrast, the HBV DNA levels are not exactly similar to the natural courses of CHB. In clinics, some patients with HBeAg (+) (IT phase or IC phase) have a high HBV DNA level, and some patients might have a low HBV DNA level. Patients with HBeAg (−) (LR phase and HBeAg-negative hepatitis phase) are believed to have low HBV DNA levels, generally; however, a study about HBeAg (−) patients had shown that 47.5% of patients in the LR phase and 63.4% in the HBeAg (−) hepatitis phase had a high HBV DNA level (>10^4^ copies/mL) [17]. Thus, the HBV DNA level would be high or undetected in these patients, which is a fact that needs further research. This phenomenon poses a challenge to clinicians regarding whether we should recommend serum HBV DNA detection for patients with CHB. This is because undergoing serum HBV DNA detection might be a burden for poor patients, and there is insufficient evidence to convince the patient to have a serum HBV DAN detection. Therefore, we need an alternative tool to evaluate HBV DNA levels roughly, providing evidence to persuade the patient to have an HBV DNA test. HBeAg plays a crucial role in HBV infection, meaning that the high replication and high infectivity of CHB has a strong correlation with HBV DNA, according to a previous study [18]. This study aims to compare the difference between HBeAg (+) and HBeAg (−) patients, and to find an alternative tool to evaluate the HBV DNA level, roughly.

## 2. Materials and Methods

### 2.1. Patients

We recruited patients from 2019 to 2020 at the Center for Liver Surgery and Liver Transplantation, West China Hospital, Sichuan University. The criteria are as follows: (1) age > 18 years, (2) return a positive HBV DNA test, (3) have HBsAg and be HBeAg positive, or have a serum HBV test return a positive result for HBsAg only. Patients who met one of the following criteria were excluded: (1) co-infection with other hepatitis viruses, such as hepatitis C, A and D; (2) having acute hepatitis, especially acute liver failure; (3) having undergone an antiviral treatment. According to previous studies [8,9,12], high HBV DNA levels were defined as >2000 IU/mL.

### 2.2. Methods

This study was approved by the West China Hospital Ethics Committee and is in accordance with the ethical guidelines of the Declaration of Helsinki. All patients have been informed about the content of the survey and they agreed the content.

We divided the patients into four groups as follows: the HBeAg (+) group, the HBeAg (−) group, the high HBV DNA level group and the low HBV DNA level group. Further, we performed a subgroup analysis for the HBeAg (+) group and the HBeAg (−) group.

### 2.3. Statistical Analysis

All data were analyzed using SPSS 22.0 data statistical software (SPSS Inc., Chicago, IL, USA). Continuous variables are expressed as mean ± standard deviation ($\bar{x}$ ± sd) values. Between-group comparisons of the continuous variables were compared by T-test or W-test, and the optimal predictive cut-off value was determined by Receiver Operating Characteristics (ROC) curves. A multivariate logistic regression analysis was used to identify the independent risk factors and to build a predictive model. The categorical variables were analyzed through chi-square test (χ^2^). For all analyses, a *p* value < 0.05 was considered statistically significant.

## 3. Results

### 3.1. The Characteristic of HBeAg (+) and HBeAg (−) Patients

We enrolled 1020 patients in this study from January 2019 to December 2020, including 881 males and 139 females, and 252 patients were HBeAg (+) and 768 were HBeAg (−). 535 patients had a high HBV DNA level and 485 patients had a low HBV DNA level. The characteristics of the patients are shown in Table 1. The values of age, the platelet count (PLT), aspartate aminotransferase (AST), and high HBV DNA levels are significant between the patients that were HBeAg (+) and HBeAg (−). The HBeAg (+) patients are younger, have a lower PLT level, and a higher AST level than the HBeAg (−) patients, and the difference is significant. In contrast, the white blood cell (WBC) count and the levels of hemoglobin (HGB), total bilirubin (TB), alanine aminotransferase (ALT), albumin (ALB), and prothrombin time (PT) are not significance. From Figure 1A, diagnosing the high HBV DNA level just based on being HBeAg (+) might be unreliable, as the AUC (area of under the curve) is 0.622, and the sensitivity is 0.557, the specificity is 0.749, positive predictive value (PPV) is 76.985% and negative predictive value (NPV) is 55.899%.

### 3.2. The Characteristics in Different HBV DNA Levels

Table 1 shows the features of high HBV DNA level and low HBV DNA level. The differences of PLT, AST, ALT, and ALB are significant between the high HBV DNA group and the low HBV DNA group, and the ROC is shown in Figure 1B. The AUC of AST, ALT, ALB, and PLT are 0.635, 0.642, 0.432, and 0.473, respectively, and the optimal cut-off values are 46.5 IU/L, 42.5 IU/L, 25.5 g/L, and 74.5 × 10^9^/L, respectively. The level of AST and ALT are higher in the high HBV DNA group, and the level of PLT and ALB are lower in this group. Furthermore, the multivariate logistic regression analysis showed no significance for PLT and AST. By combining a multivariate logistic regression analysis and univariate analysis, we find that the independent risk factors in evaluating a high HBV DNA level are ALT, ALB and HBeAg (Table 2). We propose the following predictive model for high HBV DNA levels in CHB as:Y_1_ = 0.004 × ALT (IU/L) +1.412 × HBeAg (1 for positive or 0 for negative) − 0.029 × ALB (g/L) + 0.779

The ROC of the predictive model Y_1_ is shown in Figure 2A. The AUC is 0.672, and the cut-off value is −0.072, where the sensitivity is 0.615, the specificity is 0.648, PPV is 65.182% and NPV is 60.837%.

### 3.3. The Different HBV DNA Levels in the HBeAg (+) Group

The comparison of the HBeAg (+) patients with high and low HBV DNA levels is shown in Table 3. The variables of sex, age, PLT, WBC, TB, AST, and ALT are not significantly different. The levels of HGB, ALB, and HBeAg are significant in the univariate and multivariate analysis, and the AUC is 0.394, 0.379, and 0.787, respectively, with the optimal cut-off values of 170.5 g/L, 25.0 g/L, and 16.15 S/CO, respectively, (Figure 1C). We propose that the predictive model for high HBV DNA levels in the HBeAg (+) group is the following:Y_2_ = 0.007 × HBeAg (S/CO) − 0.016 × HGB (g/L) + 3.070

The ROC of predictive model Y_2_ is shown in Figure 2C. The AUC is 0.724, and the cut-off value is 1.216, where the sensitivity is 0.626, the specificity is 0.897, PPV is 94.118% and NPV is 34.437%.

### 3.4. The Different HBV DNA Levels in the HBeAg (−) Group

Table 3 summarizes the HBV DNA level characteristics of CHB in HBeAg (−) patients. The variables of TB, AST, and ALT are significantly different between these two groups as per the univariate analysis. The AUC of these three variables are 0.511, 0.628, and 0.655, respectively (Figure 1D), and the cutoff values are 11.15 umol/L, 36.5 U/L, and 42.5 U/L, respectively. However, in the multivariate logistic regression analysis, just TB and ALT show a significant difference, as shown in Table 3. Following the abovementioned sections, we propose another predictive model for the low HBV DNA levels in the HBeAg (−) patients:Y_3_ = −0.005 × ALT(IU/L) + 0.006 × TB (umol/L) + 0.385

The ROC of the predictive model Y_3_ is shown in Figure 2B. The AUC is 0.661, and the cut-off value is 0.263, where the sensitivity is 0.677, the specificity is 0.587, PPV is 66.820% and NPV is 40.774%.

## 4. Discussion

HBV DNA is a marker of the viral activity period in CHB patients, and this supports the efficacy of antiviral therapy. Different stages of the natural progression of CHB have different characteristics, respectively, and we cannot directly judge HBV DNA levels by these characteristics [17]. This is because the existing studies suggest a weak or nonexistent correlation between HBsAg and HBV DNA levels [12,13,14,15,16], and to our best knowledge, few studies have assessed the correlation between HBeAg and HBV DNA levels. A study conducted by Chen Ping and Xie Qinfen et al. [19] showed that HBV DNA levels were at their highest (>10^7^ copies/mL) when the HbeAg levels were higher than 768 S/CO, which could indicate the relationship between HBV DNA levels and IT phages. However, they did not analyze the HBV DNA levels in HBeAg (−) patients. In this study, we analyzed the entire course of chronic hepatitis B in clinical patients. The HBeAg (+) patients were younger than HBeAg (−) patients were (*p* < 0.001, Table 1), but there was no significant difference in age between the high- and low-HBV DNA groups (*p* = 0.394, Table 1). Patients at different ages at the time of infection showed HBeAg-related behavioral differences. The existing research has showed that when patients become infected at birth or 1–2 years of age, their IC stage is prolonged [20]. In contrast, after infection in early childhood, the patients generally did not go through the IT phase and they would enter the LR phase quickly, however, this did not imply a low level of serum HBV DNA. Therefore, age is not a reasonable factor for predicting HBV DNA levels in CHB patients.

The PLT count is different between the HBeAg (+) group and the HBeAg (−) group as well as the high HBV DNA level group and the low HBV DNA level group (p_1_ = 0.001 and p_2_ = 0.011, Table 1). However, in the subgroup analysis there was no significant difference, (p_1_ = 0.739 and p_2_ = 0.086, Table 2) and previous studies have suggested a link between the PLT count and immune control of HBV infection [21,22]. By analyzing the whole natural progression of CHB, we could divide the patients into two situations: (1) the PLT count in HBeAg (+) and HBeAg (−) patients; (2) the PLT count in the high HBV DNA level group and the low HBV DNA level group. We find that the PLT is significantly difference between HBeAg (+) patients versus HBeAg (−) patients (121.6 vs. 140.81, *p* = 0.001, Table 1), and the high HBV DNA level group versus the low HBV DNA level group (129.84 vs. 142.87, *p* = 0.011, Table 1); however, the level of the PLT count is in a normal range, which does not convey useful information to identify the level of serum HBV DNA for clinicians. In previous studies, authors have reported that the level of PLT was correlated with the amount of inflammation [23,24]. A high HBV DNA level means that since the patient is in an inflammatory phase, their PLT level is lower than that of the low HBV DNA level. However, the multivariate analysis also has eliminated the possibility that the PLT count can predict the HBV DNA level. Therefore, the correlation between PLT and HBV DNA levels needs further study.

ALT level is a very important factor for CHB patients because it is a marker of liver damage. The identification of the natural progression of CHB is based on biochemical, serological, and virological characteristics, including serum ALT levels, HBeAg serostatus, and HBV DNA level [5,6,7,8]. Several studies have shown that HBV DNA levels need to be measured despite when patients have normal ALT levels [9,25]. On the other hand, high levels of ALT may be associated with HBV replication throughout chronic HBV infection, and this can be harmful to the liver [26]. Combining it with a multivariate analysis, the ALT level is an independent risk factor for predicting HBV DNA level. However, the odds ratios (OR) of the CHB group and the HBeAg (−) group are 1.004 and 1.005, respectively (Table 3), indicating that the correlation between ALT level and HBV DNA level is weak, especially in the HBeAg (+) group, which may be because HBeAg is a strong correlation of HBV DNA levels (OR = 4.104, Table 3). On the other hand, the ALT level is credible as a risk factor for predicting HBV DNA levels in the HBeAg (−) group because the AUC is 0.655 (Figure 1D). Meanwhile, there several factors would affect the level of ALT. Studies have reported that the ALT level may be varied with body mass index, abnormal lipid and carbohydrate metabolism, and the time of the day [27,28]. Thus, we need to pay attention to these factors while evaluating the HBV DNA level of HBeAg (−) patients. Generally, AST was also could be as a marker reflecting liver damage. A previous study has reported that the level of AST could reflect the HBV DNA level [29], however, it wasn’t the independent risk factor in predicting the HBV DNA level in this study and it needed further research.

Existing studies have reported weak or nonexistent correlations between HBsAg levels and HBV DNA levels [10,11,12,13,14,15]. The serum HBsAg level in the HBeAg (+) group was higher than that in the HBeAg (−) group, but as is reported in the existing studies, although HBsAg levels that are higher than 3000 IU/mL can be used as a reference for predicting high HBV DNA levels, it may be inconvenient for clinicians [7,9]. HBeAg may be a marker of high replication and high infectivity for CHB. In this study, we find that the prediction of high HBV DNA levels when they are based on HBeAg levels is reliable, as the AUC is 0.622 (Figure 1A). When the HBeAg level is higher than 16.15 S/CO, it is four times more likely that the patients will have HBV DNA levels that are higher than those who are not have lower HBeAg levels. However, following the predictive model Y_1_ = 0.004 × ALT (IU/L) + 1.412 × HBeAg (S/CO) − 0.029 × ALB (g/L) + 0.779, the AUC is 0.672 and the cut-off value is −0.072. With this we can explain that a patient with HBeAg (+) is likely to have a high level of serum HBV DNA, which is similar to whether we judged the levels of HBV DNA by their HBeAg (+) status (Figure 1A). But the AUC is higher in the model Y_1_, meaning the model Y1 is more suitable to estimate the HBV DNA levels in CHB patents rather than those that are just based on HBeAg level. On the other hand, the patients with HBeAg (−) had higher levels of HBV DNA. Therefore, we should analyze HBV DNA levels in HBeAg (+) and HBeAg (−) separately, and we conclude that HBeAg (+) is a very important risk factor for CHB patients with high HBV DNA levels.

In addition, because HBeAg (+) patients had lower HBV DNA levels, we have analyzed the different factors between the HBV DNA levels and propose a prediction model (Y_2_ = 0.007 × HBeAg (S/CO) − 0.016 × HGB (g/L) + 3.070, AUC = 0.742, and the best cut-off value is 1.216). According to predictive model Y_2_, if a patient has a level of 115 g/L of HGB, then the patient would have a high level of HBV DNA and need an HBV DNA test if they were in the HBeAg (+) group. However, from Figure 1C, we find that when the HBeAg level is higher than 16.15 S/CO in the HBeAg (+) group (the AUC is 0.787), that we should take the HBeAg level into consideration. In a word, when a patient is HBeAg (+), we should combine the level of HGB and HBeAg to estimate their HBV DNA level. On the other hand, HBeAg (−) is usually correlated with lower intrahepatic cccDNA levels [30,31,32]. Traditionally, the serum HBV DNA levels are different between HBeAg (+) and HBeAg (−) patients. However, Lai CL and Ratziu V et al. reported that HBeAg (−) patients did not have low HBV DNA levels [20]. Following a multivariate logistic regression analysis, the independent risk factors to predict the serum HBV DNA in HBeAg (−) are the levels of TB and ALT, and the cut-off values are 11.15 umol/L and 36.5 IU/L, respectively (Table 2). We propose the predictive model Y_3_ (Y_3_ = −0.005 × ALT(IU/L) + 0.006 × TB (umol/L) + 0.385, AUC = 0.661, the cut-off value = 0.263) for the patients that are HBeAg (−). We find that when the levels of TB and ALT are higher than 11.15 umol/L and 36.5 IU/L, respectively, then these patients should be requested to have a HBV DNA detection. Certainly, HBV infection and its reproduction occurs throughout all phages, thus, we cannot ignore the level of HBV DNA in HBeAg (−) patients.

Last but not least, the present study has several limitations: (i) Cross-sectional studies just reflect the present situation, so a long-term follow-up may be needed to identify the parameters that reflect the responses of patients who have received the antivirus treatment, especially those who had high levels of HBV DNA. In the current study, we did not perform a follow-up to explore the effects of the antiviral treatment. (ii) We did not divide the patients into different phases of CHB because it was difficult to clarify the phases of CHB in this study. (iii) The level of HBV DNA may be different between HBV genotype A and D, and we did not assess the HBV genotype in these patients. However, to our best knowledge, this study is the first research study focusing on the clinical parameters for evaluating HBV DNA levels to predict HBV activity. Although the HbeAg status was negative in some patients, these patients were needed to detect their HBV DNA level [33]. In our findings, we provide a convenient method to estimate the effect of an antivirus therapy, which might reduce the patient’s costs (i.e., the cost of HBV DNA test is about 10 dollars in China).

## 5. Conclusions

HBeAg is an independent risk factor that reflects the level of HBV DNA with a strong correlation. Patients with HBeAg (−) should combine TB and ALT to estimate their level of HBV DNA.

## Figures and Tables

**Figure 1 medicina-58-01279-f001:**
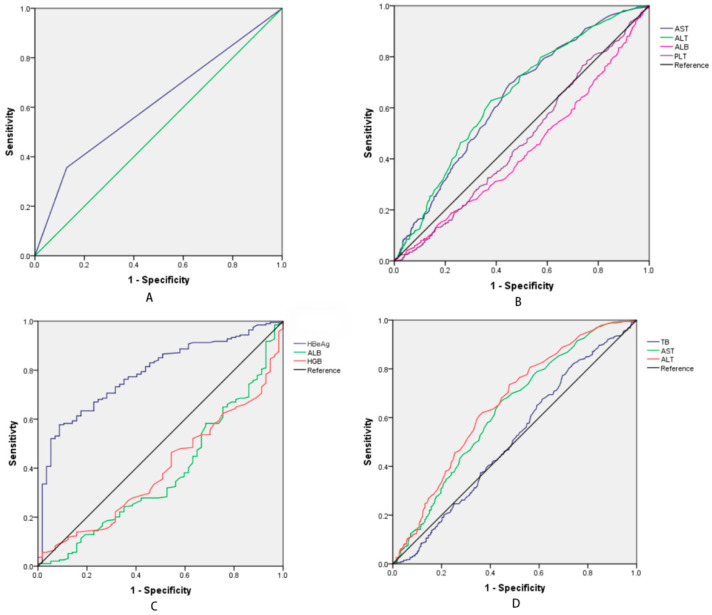
(**A**) The ROC of judging the HBV DNA level by the HBeAg (+) status of the patients with CHB, and the AUC is 0.622. (**B**) The ROC of significant factors for different levels of HBV DNA. These factors are AST, ALT, ALB and PLT, and the AUC 0.635, 0.642, 0.432 and 0.473, respectively. The best cut-off values are 46.5 IU/L, 42.5 IU/L, 25.5 IU/L and 74.5 × 10^9/L, respectively. (**C**) The ROC of HbeAg, ALB and HGB in HbeAg (+) group, and the ACU of HbeAg, ALB and HB are 0.787, 0.379 and 0.394, respectively. The best cut-off value of HBeAg is 16.15 S/CO. (**D**) The ROC of TB, AST and ALT in HBeAg (−) group, the AUC and cut-off values are 0.511, 0.628, 0.655 and 11.15 umol/L, 42.5 IU/L, 36.5 IU/L, respectively.

**Figure 2 medicina-58-01279-f002:**
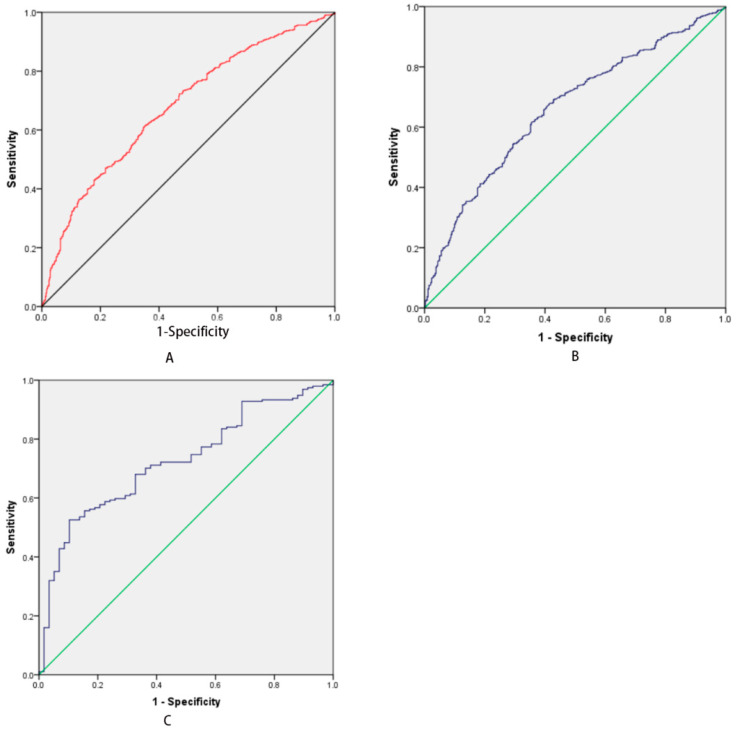
(**A**) Different HBV DNA levels in CHB patients. The ROC of the predictive model Y_1_ (high HBV DNA level in CHB patients), the ACU and best cut-off values are 0.672 and 0.072, respectively. (**B**) HBV DNA levels in HBeAg (−) patients. The ROC of the predictive model Y_3_ (low HBV DNA level in HBeAg (−) patients), the AUC and cut-off values are 0.661 and 0.263, respectively. (**C**) HBV DNA level in HBeAg (+) patients. The ROC of predictive model Y_2_ (high HBV DNA level in HBeAg (+) patients), the AUC and cut-off values are 0.742 and 1.216, respectively.

**Table 1 medicina-58-01279-t001:** The characteristic features of the natural course of CHB in patients.

Variable	HBeAg			HBV DNA Levels		
	HBeAg (+)	HBeAg (−)	P1 Value	High	Low	P2 Value
Sex (male/female)	223/29	658/110	0.258	474/61	407/78	0.030
Age (years)	46.49 ± 10.47	52.12 ± 12.03	<0.001	50.42 ± 11.29	51.05 ± 12.56	0.394
HGB (g/L)	135.63 ± 23.00	134.90 ± 21.69	0.650	134.46 ± 22.88	135.78 ± 21.02	0.340
PLT (×10^9^/L)	121.60 ± 76.57	140.81 ± 82.00	0.001	129.84 ± 70.75	142.87 ± 90.80	0.011
WBC (×10^9^/L)	5.48 ± 3.38	5.91 ± 3.42	0.083	5.66 ± 3.23	5.95 ± 3.60	0.184
TB (umol/L)	25.94 ± 53.78	25.37 ± 51.48	0.878	23.18 ± 45.26	28.09 ± 58.7	0.133
AST (IU/L) (Median)	96	65	0.023	100	60	0.001
ALT (IU/L) (Median)	72	51	0.071	85	44	<0.001
ALB (g/L)	38.42 ± 7.51	39.21 ± 5.63	0.080	38.46 ± 6.56	39.62 ± 5.63	0.003
PT (s)	12.87 ± 2.35	12.71 ± 2.30	0.362	12.84 ± 2.19	12.65 ± 2.44	0.177
HBV-DNA levels (>2 × 10^3^ IU/mL × 10^3^ IU/mL)	194/58	341/427	<0.001	-	-	-
HBeAg (+)/HBeAg (−)	-	-	-	194/341	58/427	<0.001

HBeAg: Hepatitis B ‘e’ antigen, HGB: Hemoglobin, PLT: Platelet, WBC: White blood cell, TB: Total bilirubin, AST: Aspartate aminotransferase, ALT: Alanine aminotransferase, ALB: Albumin, PT: Prothrombin time.

**Table 2 medicina-58-01279-t002:** The results of the logistic regression multivariate analysis.

CHB					
Variable	B	*p*	Exp(B)	Exp(B) 95% CI	
				Down	Up
ALT	0.004	0.002	1.004	1.002	1.007
ALB	−0.029	0.010	0.971	0.950	0.993
HBeAg (+)	1.412	<0.001	4.104	2.941	5.726
Constant	0.779	0.035	2.178	-	-
HBeAg (+)					
HGB	−0.016	0.028	0.984	0.970	0.998
HBeAg	0.007	0.011	1.007	1.001	1.012
Constant	3.070	0.003	21.542	-	-
HBeAg (−)					
TB	−0.006	0.008	0.994	0.989	0.998
ALT	0.005	<0.001	1.005	1.003	1.008
Constant	−0.385	0.001	0.680	-	-

CHB: Chronic Hepatitis B, HBeAg (−): Hepatitis B ‘e’ antigen negative, B: Partial regression coefficient value, Exp (B): Odds Ratio (OR), CI: confidence interval.

**Table 3 medicina-58-01279-t003:** Comparison of HBV DNA levels between HBeAg (+) and HBeAg (−) patients.

	HBeAg (+)				HBeAg (−)			
	High HBV DNA	Low HBV DNA	p_1_ Value	Cut-Off Value	High HBV DNA	Low HBV DNA	p_2_ Value	Cut-Off Value
Sex (male/female)	172/22	51/7	0.879	-	302/49	356/71	0.305	-
Age (years)	46.82 ± 10.58	45.44 ± 9.93	0.378	-	52.46 ± 11.18	51.82 ± 12.7	0.463	-
HGB (g/L)	133.71 ± 23.94	142.09 ± 18.19	0.015	170.5	134.89 ± 22.27	134.93 ± 21.25	0.979	-
PLT (×10^9^/L)	120.67 ± 72.38	124.53 ± 90.49	0.739	-	135.09 ± 69.37	145.36 ± 90.66	0.086	-
WBC (×10^9^/L)	5.47 ± 3.28	5.52 ± 3.74	0.912	-	5.78 ± 3.19	6.00 ± 3.59	0.359	-
TB (umol/L)	27.51 ± 60.89	21.04 ± 15.67	0.425	-	20.71 ± 60.89	29.05 ± 62.12	0.026	11.15
AST (IU/L)	101.48 ± 141.86	73.78 ± 106.62	0.170	-	77.81 ± 85.51	64.13 ± 79.50	0.022	42.5
ALT (IU/L)	71.29 ± 95.64	60.36 ± 59.75	0.411	-	68.10 ± 64.78	52.00 ± 59.54	0.001	36.5
ALB (g/L)	37.88 ± 7.76	40.23 ± 6.37	0.036	25.0	38.79 ± 5.75	39.54 ± 5.52	0.068	-
PT (s)	12.98 ± 2.52	12.49 ± 1.62	0.173	-	12.77 ± 1.96	12.67 ± 2.53	0.556	-
HBeAg levels (S/CO)	138.00 ± 238.74	24.46 ± 132.97	0.001	16.15	-	-	-	-

## Data Availability

The data sets used during the current study are available from the corresponding author on reasonable request.

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
