# Peer review of "A Predictive Model to Evaluate the HbeAg Positivity of Chronic Hepatitis B Virus Patients in Clinics: A Cross-Sectional Study"

_medicina, 2022, doi:10.3390/medicina58091279_

Round 1

Reviewer 1 Report

This study is about adopting several biochemical tests to predict HBV DNA using a formula derived from logistic regression. The cutoff of the equation is derived from ROC. The intention is to save cost on unnecessary HBV DNA testing. Although AUC is shown, it would be better to show the classification accuracy, sensitivity, specificity, PPV and NPV so that more readers would appreciate the usefulness of the model/equation.

Similar paper with similar results was published in research square with different authors!! [https://assets.researchsquare.com/files/rs-151954/v1/3883e5f7-4f7e-4c05-b7c9-1708cd8db041.pdf?c=1631871853]

No mention of ethical approval and sample size in methodology.

Too many grammar and spelling errors. And rephrase of many sentences needs to be done. Manuscript writing was not careful and of low quality.

Author Response

Dear Editor:

Thank you for your letter on August, 24, 2022 and your attention to our manuscript (Title: A predicted model to evaluate the HbeAg-positive of Chronic Hepatitis B Virus in clinic: a cross-sectional study). We thank both editors and reviewers for their positive and constructive comments and suggestions.

We have revised the manuscript carefully, according to the comments and suggestions of editors and reviewers, and responded, point by point, the comments as listed below. According to your request, we upload a supplementary file of a version of the manuscript with the changes highlighted and another unmarked revised manuscript.

We would like to re-submit this revised manuscript to Medicina, and hope it is acceptable for publication in the journal.

Looking forward to hearing from you soon.

With kindest regards,

Yours Sincerely

Li Jiang, M.D.

Department of Liver Surgery,

Liver Transplantation Center,

West China Hospital of Sichuan University

Response to Editors and Reviewers

First of all, we thank both editors and reviewers for their constructive comments and suggestions, which are very valuable for improving the quality of our manuscript.

Response to comment of reviewer 1:

Q 1.This study is about adopting several biochemical tests to predict HBV DNA using a formula derived from logistic regression. The cutoff of the equation is derived from ROC. The intention is to save cost on unnecessary HBV DNA testing. Although AUC is shown, it would be better to show the classification accuracy, sensitivity, specificity, PPV and NPV so that more readers would appreciate the usefulness of the model/equation.

Answer:We have added these parameters.(sensitivity, specificity, PPV and NPV )

Q 2. Similar paper with similar results was published in research square with different authors!! [https://assets.researchsquare.com/files/rs-151954/v1/3883e5f7-4f7e-4c05-b7c9-1708cd8db041.pdf?c=1631871853]

Answer: The article was published in square with different authors because we did a few modifications and we also communicated with the relevant authors and obtained their consent.

Q 3. No mention of ethical approval and sample size in methodology.

Answer:We have added these complements in methodology.(line 5-7, page 5)

Q 4. Too many grammar and spelling errors. And rephrase of many sentences needs to be done. Manuscript writing was not careful and of low quality.

Answer:We have completed the language editing by www. enago. cn.

Reviewer 2 Report

Dear editor

Thanks for inviting me to review the manuscript medicina-1807681 entitled "A predicted model to evaluate the HbeAg-positive of Chronic Hepatitis B in clinic: a cross-sectional study".

I appreciate the authors' work to collect these data from 1020 patients. Although the subject is not novel, these data of 1020 patients are reportable. My comments to the authors are as follow.

Abstract:

- Please use the extended forms of the abbreviations for the first time in this section.

- Please use further numeric results in this section (e.g. P value, cut-off points, etc.).

- In the conclusion section, what do you mean of "strong correlation"?

Introduction:

- This section needs some recent studies (2018-2022) to show the importance of the subject. 

- Please use the extended forms of the abbreviations for the first time in this section.

Methods:

- This section needs further details regarding the laboratory techniques.

- Did you evaluate the normality of the distributions?

- Please mention the univariate analyses to compare the groups (in both normal/not-normal distribution conditions).

Results:

- Please show the reference line in all of the ROC-Curve figures.

- Please mention the measured specificity and sensitivity values for all cut-off points.

- Please use three decimals for all P values.

Discussion:

- This section needs to be updated using recently published articles.

- Please discuss the importance of this study.

References:

- Some references seem outdated and need to be changed (e.g. NO.26, Hepatology 1998)

- This manuscript needs some recent studies (2018-2022).

Regards

Author Response

Response to comment of reviewer 2:

Abstract:

- Please use the extended forms of the abbreviations for the first time in this section.

Answer:We have used the extended forms of the abbreviations for the first time.

- Please use further numeric results in this section (e.g. P value, cut-off points, etc.).

Answer:We have used the numeric results in abstract.

- In the conclusion section, what do you mean of "strong correlation"?

Answer:We have corrected this part.

Introduction:

- This section needs some recent studies (2018-2022) to show the importance of the subject. 

Answer:We have added several recent studies in this part. (line 7-10, page 3, NO 4 and 5)

- Please use the extended forms of the abbreviations for the first time in this section.

Answer:We have used the extended forms of the abbreviations for the first time.

Methods:

- This section needs further details regarding the laboratory techniques.

Answer:We have added the details in this part.(line 5-7, page 5)

- Did you evaluate the normality of the distributions?

Answer:We have done the normality of distributions and have corrected the method of statistical analysis.(table 1)

- Please mention the univariate analyses to compare the groups (in both normal/not-normal distribution conditions).

Answer:We have corrected the method of statistical analysis.(table 1)

Results:

- Please show the reference line in all of the ROC-Curve figures.

Answer:We have added the reference line in ROC curve figures. (figure 1 and figure 2)

- Please mention the measured specificity and sensitivity values for all cut-off points.

Answer:We have added the specificity and sensitivity values for all cut-off points.

- Please use three decimals for all P values.

Answer:We have used decimals for all p values.(table 1)

Discussion:

- This section needs to be updated using recently published articles.

Answer:We have added several recent studies in this part.

- Please discuss the importance of this study.

Answer:We have added the discussion in this study.

References:

- Some references seem outdated and need to be changed (e.g. NO.26, Hepatology 1998)

Answer:We have changed these references. (NO 30)

- This manuscript needs some recent studies (2018-2022).

Answer:We have added some recent studies.

Reviewer 3 Report

The manuscript submitted by Ning Wang et al. entitled “A predicted model to evaluate the HbeAg-positive of Chronic Hepatitis B in clinic: a cross-sectional study”, proposed to investigate the correlation between HBeAg and HBV DNA levels and find an alternate tool to evaluate HBV DNA levels in CHB patients.

Major strengths, concerns and limitations of the manuscript: The authors addressed the correlations among CHB patients based on HBeAg status and HBV DNA levels to sort out to estimate serum HBV DNA levels in patients. They enrolled good number of patients for this study, assessed various characteristic features responsible for the natural course of CHB infection in patients. The authors provided reasonable amount of data to support the study with many limitations. The authors conclude that HbeAg status is an independent risk factor that reflects the level of HBV DNA levels with strong correlation. HbeAg (-) patient should combine TB and ALT to estimate the levels of HBV DNA in patients. Although the study may not be novel, but this is an interesting study to support the clinical testing in multiple CHB patients based on HBeAg status if author provides some additional data for alternate tool to investigate HBV DNA levels and carry out major corrections as detailed below.

Major concerns need to be addressed:

1.     Need extensive typographical error corrections throughout the manuscript.

2.     In abstract, Line 11: change the word we “discovery” to we “propose”.

3.     In introduction part: Line 4, correct over 350 million people facing been affected or have been affected? Before end of introduction section, change the word “these two style patients”.

4.     In result part 3.1 section - correct “585 patients with a low HBV DNA level” to “485 patients”.  Section 3.3: in Line 3, “ALB” not ALT. Section 3.4: in line 1, add Table 2 “also” summarizes the HBV DNA levels. Line 5 end, change text to “only TB and ALT are significantly different as shown in Table 3”.

5.     In Table 2: no of HBeAg (-) patients “768” not “778” in total. Need correction in HBV DNA level patient’s total number?

6.     Figures are poor in presentation, barely visible labeling and provide a high-quality figures.

7.     Re-arrange the tables according to the results. Follow one format labeling like Figure 1, 2 and Table 1, 2,3…or figure 1,2 and table 1,2,3…change words “cutoff” to ”cut-off” and find to found wherever applicable.

8.     Corrections in Figure 2. A description - HBV “DNA” not HBV DAN, values not value.

9.     Add “characteristics” and “levels” wherever appropriate.

10.  Modify the titles for Table 1 as “The characteristic features of the natural course of CHB in patients”. For Table 2 as “Comparison of HBV DNA levels between HBeAg (+) and HBeAg (-) patients”.

11.  Can you find reasons with additional experiment / data or give more clarification on the correlation between low PLT vs HBV DNA levels?

12.  Give reasons for correlation between AST x HBV DNA levels?

13.  Improve discussion part correlating with results and correct typo errors like “phases” not “Phages”.

Author Response

Response to comment of reviewer 3:

  1. Need extensive typographical error corrections throughout the manuscript.

Answer:We have completed the language editing by www. enago. cn.

  1. In abstract, Line 11: change the word we “discovery” to we “propose”.

Answer:We have corrected this word.

  1. In introduction part: Line 4, correct over 350 million people facing been affected or have been affected? Before end of introduction section, change the word “these two style patients”.

Answer:We have corrected this part.

  1. In result part 3.1 section - correct “585 patients with a low HBV DNA level” to “485 patients”.  Section 3.3: in Line 3, “ALB” not ALT. Section 3.4: in line 1, add Table 2 “also” summarizes the HBV DNA levels. Line 5 end, change text to “only TB and ALT are significantly different as shown in Table 3”.

Answer:We have corrected these mistakes.

  1. In Table 2: no of HBeAg (-) patients “768” not “778” in total. Need correction in HBV DNA level patient’s total number?

Answer:We have corrected this mistakes and the number of HBeAg(-) patient was 778.

  1. Figures are poor in presentation, barely visible labeling and provide a high-quality figures.

Answer:We have uploaded the new figures

  1. Re-arrange the tables according to the results. Follow one format labeling like Figure 1, 2 and Table 1, 2,3…or figure 1,2 and table 1,2,3…change words “cutoff” to ”cut-off” and find to found wherever applicable.

Answer:We have re-arranged the tables according to the results.

  1. Corrections in Figure 2. A description - HBV “DNA” not HBV DAN, values not value.

Answer:We have corrected this mistakes.

  1. Add “characteristics” and “levels” wherever appropriate.

Answer:We have added these words appropriate.

  1. Modify the titles for Table 1 as “The characteristic features of the natural course of CHB in patients”. For Table 2 as “Comparison of HBV DNA levels between HBeAg (+) and HBeAg (-) patients”.

Answer:We have completed this change.

  1. Can you find reasons with additional experiment / data or give more clarification on the correlation between low PLT vs HBV DNA levels?

Answer:We have completed this change.

  1. Give reasons for correlation between AST x HBV DNA levels?

Answer:We have explained the correlation between ALT x HBV DNA levels, however, the AST wasn’t the risk factor for HBV DNA.

  1. Improve discussion part correlating with results and correct typo errors like “phases” not “Phages”.

Answer:We have corrected this mistakes.

Round 2

Reviewer 1 Report

Line numbering is absent. Difficult to review.

Lack of rigors in scientific writing.

Page 2: "... more than 350 million people are newly infected each year ..." 350 million is an outdated data. Please refer to latest reference regarding this.

Page 3: "it is insufficient evidence to convince the patient to have HBV DAN test." What is HBV DAN test?

Page 4: Inappropriate ethical statement.

"And all patients have been informed content of survey." did not indicate that the consent of the study had been obtained from the subjects.

Author Response

Line numbering is absent. Difficult to review.

Answer: We have added the numbering in the manuscript.

Lack of rigors in scientific writing.

Answer: We have edited the writing.

Page 2: "... more than 350 million people are newly infected each year ..." 350 million is an outdated data. Please refer to latest reference regarding this.

Answer: We have updated this new data in reference.

Page 3: "it is insufficient evidence to convince the patient to have HBV DAN test." What is HBV DAN test?

Answer: We have updated this new data in reference 1.

Page 4: Inappropriate ethical statement.

"And all patients have been informed content of survey." did not indicate that the consent of the study had been obtained from the subjects.

Answer: We have added the statement in page 5 (line 122-123)

Reviewer 2 Report

Dear Editor

Thanks for inviting me to review the revised version of the manuscript medicina-1807681 entitled "A predicted model to evaluate the HbeAg-positive of Chronic Hepatitis B in clinic: a cross-sectional study".

I appreciate the authors' work to revise their manuscript. All of my comments are addressed by the authors, and I think this version of the manuscript is good enough to be published in the Medicina Journal.

Regards

Author Response

Thanks for inviting me to review the revised version of the manuscript medicina-1807681 entitled "A predicted model to evaluate the HbeAg-positive of Chronic Hepatitis B in clinic: a cross-sectional study".

I appreciate the authors' work to revise their manuscript. All of my comments are addressed by the authors, and I think this version of the manuscript is good enough to be published in the Medicina Journal.

Reviewer 3 Report

Manuscript can be improved with English corrections and spell checks.

Author Response

Manuscript can be improved with English corrections and spell checks.

Answer: We have edited the writing.